# The Role of Oxidative Stress in the Risk of Cardiovascular Disease and Identification of Risk Factors Using AIP and Castelli Atherogenicity Indicators in Patients with PCOS

**DOI:** 10.3390/biomedicines10071700

**Published:** 2022-07-14

**Authors:** Jolanta Nawrocka-Rutkowska, Iwona Szydłowska, Katarzyna Jakubowska, Maria Olszewska, Dariusz Chlubek, Małgorzata Szczuko, Andrzej Starczewski

**Affiliations:** 1Department of Gynecology, Endocrinology and Gynecological Oncology, Pomeranian Medical University, Unii Lubelskiej Street 1, 71-252 Szczecin, Poland; iwona.szydlowska@pum.edu.pl (I.S.); andrzejstarcz@o2.pl (A.S.); 2Department of Biochemistry and Medical Chemistry, Pomeranian Medical University, Powstańców Wielkopolskich Street 72, 70-111 Szczecin, Poland; katarzyna.jakubowska@pum.edu.pl (K.J.); maria.olszewska@pum.edu.pl (M.O.); dariusz.chlubek@pum.edu.pl (D.C.); 3Department of Human Nutrition and Metabolomic, Pomeranian Medical University in Szczecin, 71-460 Szczecin, Poland; malgorzata.szczuko@pum.edu.pl

**Keywords:** PCOS, oxidative stress, atherogenicity indicators, AIP, Castelli index, cardiovascular disease

## Abstract

Polycystic ovarian syndrome (PCOS) is one of the most common endocrinopathies in females of reproductive age and may affect 5–14% of women. In women with PCO syndrome, metabolic disorders such as insulin resistance, hyperinsulinemia, obesity, diabetes mellitus, and other elements of metabolic syndrome may occur. Patients with PCOS often have overweight and obesity, especially abdominal obesity, which is one of the risk factors for developing atherosclerosis. The atherogenicity indicators of AIP (atherogenic index of plasma) and Castelli’s index are used to assess the risk of developing atherosclerosis. Studies have shown an increase in the concentration and activity of oxidative stress markers in patients with PCOS compared to women without the disease. The aim of the present study was to evaluate oxidative stress parameters in patients with PCOS in relation to insulin resistance, BMI, and hyperandrogenemia and to correlate them with cardiovascular risk parameters. Conclusions: The severity of oxidative stress in women with PCOS correlates with exposure to cardiovascular diseases. The assessment of additional cardiovascular disease (CVD) parameters is useful in identifying the risk groups for cardiometabolic disease among PCOS patients. When additional risk factors such as hyperandrogenism and insulin resistance (IR) are present in patients with PCOS, it is reasonable to include preventive examinations early. It is also important to evaluate lipidograms, which will make it possible to determine indicators of atherogenicity. Patients with PCOS and IR are at particular risk for cardiovascular complications. PCOS should be considered an important risk factor for CVD, which occurs independently of the occurrence of obesity. This factor is related to the important role of insulin resistance, which occurs independently of obesity. Atherogenic factors (AIP and Castelli index) are useful additional parameters to assess the risk of cardiometabolic disease in PCOS patients, especially among groups with insulin resistance. The early detection of risk factors should be an integral part of the care of PCOS patients. In laboratory studies of women with PCOS, TG, TChol, HDL-c and LDL-c levels, and glutathione peroxidase (GPx) activity were most clearly correlated with exposure to cardiovascular disease.

## 1. Introduction

Polycystic ovarian syndrome (PCOS) is one of the most common endocrinopathies in females of reproductive age and may affect 5–14% of women [1,2]. At present, the etiology of this syndrome is not fully understood. In women with PCO syndrome, metabolic disorders such as hyperinsulinemia, insulin resistance (IR), obesity, non-alcoholic fatty liver disease (NAFLD), diabetes mellitus, and other elements of metabolic syndrome may occur [3].

### 1.1. Metabolic Syndrome

It is estimated that approximately 90% of obese women with PCOS and 75% of slim women with PCOS have coexisting metabolic disorders [4]. Nearly 50% of PCOS women progress to metabolic syndrome (MetS). MetS refers to a group of conditions, including dyslipidemia, impaired glucose tolerance, hypertension, central adiposity, and cardiovascular disease (CVD) [5,6,7]. All of these conditions can cause coronary heart disease and diabetes [8,9]. MetS is diagnosed by the presence of any three of the following: elevated triglycerides (TGs) of ≥150 mg/dL; a reduced high-density lipoprotein cholesterol fraction (HDL-c) of <50 mg/dL in women; an elevated waist circumference of ≥89 cm in women; elevated blood pressure (≥130 mmHg systolic or ≥85 mmHg diastolic); and elevated fasting glucose ≥ 100 mg/dL, waist circumference, or waist–hip ratio (WHR) [10,11]. Individual predispositions and environmental factors such as improper diet and a lack of physical activity are particularly dangerous for this group of patients. Together, these factors form a triad that predisposes patients to the occurrence of diseases of the circulatory system, including ischemic heart disease (IHD), coronary artery disease (CAD), and atherosclerosis. To determine the risk of CVD, selected MetS parameters and indicators used in atherogenicity diagnostics were analyzed.

### 1.2. Atherogenicity Diagnostics

The atherogenic index of plasma (AIP) and Castelli index are used as markers of plasma atherogenicity [12,13]. AIP is an indicator of dyslipidemia that is calculated from the concentration of the high-density lipoprotein cholesterol fraction (HDL-c) and triglyceride (TG). The Castelli index is a more accurate predictor of cardiovascular risk than other commonly used parameters such as the lipid concentration. To date, individual studies in the literature have used the AIP indicator to analyze the risk of cardiovascular disease in women with PCOS. We did not find studies that used the Castelli index.

### 1.3. Ovulation Disturbances

Excessive weight and obesity can disrupt ovulation and cause or intensify insulin resistance and hyperandrogenism. Progressive disorders associated with obesity include an increase in adipogenesis and a decrease in lipolysis. Interactions between adipose tissue and the ovaries result in abnormal folliculogenesis and may lead to the destruction of oocytes. Due to obesity, proinflammatory adipokines are secreted. Through their influence on the thecal cells, these adipokines interfere with the production of ovarian androgens [7,14]. Excessive androgen production by the thecal cells of the ovary is considered to be caused by constant acyclic stimulation via luteinizing hormone (LH), whereas a relative follicle-stimulating hormone (FSH) deficiency is responsible for a chronic lack of ovulation. Ovarian hyperandrogenism results in disorders of selection, ovarian follicular maturation, and ovulation. Thus, polycystic ovaries have a significant number of small follicles that do not ovulate. A lack of ovulation can cause menstrual disorders and infertility [15]. Hyperandrogenemia is further exacerbated by hyperinsulinemia while decreased sex hormone-binding globulin (SHBG) concentrations cause large quantities of androgens to circulate in a free biologically active form.

### 1.4. Chronic Inflammation and Oxidative Stress

Metabolic disorders in PCOS cause an increase in the number of free radicals, which cause increased oxidative stress (OS) and inflammation, which is associated with IR in PCOS [16]. A number of studies indicated the role of chronic inflammation in PCOS, which is directly related to the risk of CVD. Chronic inflammation correlates not only with obesity and hyperinsulinemia but also with an excess of androgens. Such inflammation may lead to adipocyte hypertrophy and, consequently, result in tissue hypoxia. Obesity, hyperinsulinism, IR, and OS affect abnormal glucose metabolism while high insulin concentrations resulting from insulin resistance have an impact on protein oxidation. Therefore, the use of antioxidant therapy should have a positive effect on the improvement of the insulin sensitivity of tissues. [17,18,19]. Oxidative stress results from imbalances between oxidants and antioxidants. The reactive forms of oxygen and nitrogen can cause damage to cellular lipids, proteins, and deoxyribonucleic acid (DNA) [20]. The following antioxidants occur in the body: enzymes, which include catalase (CAT), gluthation peroxidase (GPx), and superoxide dismutase (SOD); macromolecular antioxidants (i.e., malondialdehyd (MDA)); and low-molecular-weight antioxidants, including glutathion, albumin, ferritin, vitamin E, and ascorbic acid.

SOD is an enzyme that alternately catalyzes dismutation of superoxide radicals into ordinary molecular oxygen and hydrogen peroxide. SOD is an antioxidant defense in nearly all living cells exposed to oxygen [21].

CAT is another marker of oxidative stress. In arteriosclerosis and diabetes, reduced catalase activity is observed, which indicates the occurrence of long-term oxidative stress.

GPx and oxyreductive enzymes catalyze the reactions of oxidation and reduction. The former catalyzes the reactions of hydrogen peroxide reduction while the latter breaks down hydrogen peroxide. As the accumulation of peroxides may lead to the formation of free radicals, peroxidase protects the body against the harmful effects of free radicals. Peroxidase, along with vitamin E, constitutes a part of the anti-lipid oxidation protective system [22]. MDA is a marker of oxidative stress and is also used to control the effectiveness of antioxidant therapy. The concentration of MDA rises under increased production of reactive forms of oxygen (increase in the free radicals concentration). A reduction in the concentration of OS parameters can be achieved with a diet rich in antioxidants and/or a calorie-reduced diet, leading to weight and adipose tissue reduction [23,24].

Studies have shown an increase in the concentration and activity of OS markers in patients with PCOS compared to women without the disease. The aim of the present study was to evaluate oxidative stress parameters in patients with PCOS in relation to cardiovascular risk and anthropometric parameters, and the markers of plasma atherogenicity.

## 2. Materials and Methods

### 2.1. Study Group of Patients

Fifty-six premenopausal women were evaluated in this study. All patients were recruited at the Department of Endocrinology, Gynecology, and Gynecological Oncology Pomeranian Medical University in Szczecin, Poland, in 2017–2020. The study group consisted of 29 patients who were hospitalized for menstrual disorders. The Rotterdam classification was used to diagnose PCOS [25]. Hirsutism was diagnosed when the severity of symptoms was assessed as >8 points according to the Ferriman–Gallwey scale [26]. The control group consisted of 27 potentially healthy women who were recruited during screening. The exclusion criteria in both groups were: hypertension; diabetes mellitus; ischemic heart disease; therapy with insulin-sensitizing drugs, lipid-lowering drugs or antioxidant supplementation; and current or past hormonal therapy. Ultrasound examinations were carried out with a General Electric, Voluson E8 Expert, vaginal head at 7.5 Mhz.

Hirsutism was diagnosed when the severity of symptoms was assessed as >8 points. The characteristics of the study group and division into subgroups are presented in Table 1. For every patient, the body mass index (BMI) and waist–hip ratio (WHR) were measured. Based on the American Association of Clinical Endocrinologists (AACE) and American College of Endocrinology (ACE) guidelines of 2016, we accepted the following values: regular BMI values: 18.5–24.99 and obesity and overweight: ≥25 kg/m^2^ [27,28,29]. WHR >0.8 indicated abdominal obesity. The blood pressure of patients was also determined.

### 2.2. Blood Chemistry Measurements

Fasting blood samples were collected between the second and fifth day of the cycle via ulnar vein puncture. Concentration levels of testosterone (T), androstendione (A), SHBG, glucose, and insulin (I) (fasting and 120 min later after an oral load of 75 g of glucose), TG, HDL-c, LDL-c, and TChol were determined from peripheral blood serum. Two ethylendiaminetetraacetic acid (EDTA) tubes containing 7 mL of blood in total were centrifuged (1000× *g*, 4 °C, 15 min), and plasma was separated and frozen at −80 °C.

### 2.3. Analyzed Indicators

Homeostatic Model Assessment (HOMA= fasting glucose [mmol/L] × fasting insulin [μIU/mL]/22.5) was used to assess IR. IR was diagnosed when the value was equal to or greater than 3.8 [30,31].

The free androgen index (FAI) was calculated as T (nmol/L)/SHBG (nmol/L) × 100 [32].

To determine the risk of CVD, the Castelli index (atherogenicity index calculated from the ratio of TChol/HDL-c fraction value (N < 4.0; value < 3.0 after myocardial infarction; 2.5 is a very good value)) and the AIP, which seems to correlate best with the risk of ischemic heart disease, were also calculated [33,34] according to the following formula: logarithm from the quotient of the triglyceride concentration value and the HDL-c fraction concentration value. A value greater than 0.5 was taken to indicate an increased risk of cardiovascular disease. To investigate the relationship between AIP, the Castelli index, and anthropometric parameters in patients with PCOS, this group of women was divided into two subgroups depending on the concomitant IR, hyperandrogenism, and overweight and obesity as follows: with IR (*n* = 8) and without IR (*n* = 21); with hyperandrogenism (*n* = 14) and without hyperandrogenism (*n* = 15); and with overweight and obesity (*n* = 14) and normal body weight (*n* = 15). In each group, the values of the AIP and Castelli indicators were compared.

### 2.4. Analysis of Oxidative Stress Parameters

To establish the contents of hemoglobin in the tested samples of hemolysates, the Drabkin method was applied [35]. This determination must have been made before the spectrophotometric assessment of the enzymatic determination of the enzymatic activity of CAT, SOD, and GPx.

The enzymatic activity of GPx in the erythrocyte samples was determined by the spectrophotometric Wendel method [36].

MDA determination was performed in high-performance liquid chromatography (HPLC) [37].

The enzymatic CAT activity in erythrocytes was determined according to the Aebi method using a spectrophotometer [38].

The SOD activity in erythrocytes was determined by the spectrophotometric method based on the Misra and Fridovich method [39].

### 2.5. Statistical Analysis

Statistical analyses were conducted using SPSS Statistics, version 13.0 (StatSoft, Cracow, Poland). The collected data were characterized using descriptive statistics (mean, standard deviation (SD), and minimum and maximum values). The assumption about the normality of the distribution of variables was checked using a W Shapiro–Wilk test. Intergroup comparisons of quantitative variables were performed using a Mann–Whitney U test. Intergroup comparisons of qualitative variables were performed using a Chi2 test. In the correlation analysis, Spearman’s R factor was used. The results were considered statistically significant at *p* < 0.05.

## 3. Results

The mean age of the study group (*n* = 29) was 27.14 ± 7.08 while that of the control group (*n* = 27) was 35.78 ± 9.83. The difference between the groups was significant. Hirsutism of >8 points was found in 6.89% (*n* = 2) and acne in 55.1% (*n* = 16) of PCOS patients. Hirsutism was not found in the control group while acne was present in 3.7% (*n* = 1) of the control group. These differences were significant.

Insulin resistance was found in 27.6% (*n* = 8) of the PCOS group and 7.4% (*n* = 2) of the control group. Hyperandrogenism (increased androstenedione values of >3.5 ng/mL and/or hirsutism of >8 points on the Ferriman–Gallwey scale) was found in 48.3% (*n* = 14) of the study group and 7.4% (*n* = 2) of the control group. Differences were significant at *p* = 0.001. The results of the anthropometric analyses in the group with PCOS and the control group are described in Table 1. The differences between groups were not significant.

Systolic pressure values did not differ significantly between the study and control groups; the means were as follows: 118.79, SD: 14.46 vs. 110.56, SD: 14.16. Similarly, there were no significant differences in the diastolic pressure values between the study and control groups. The mean values were as follows: 78.17, SD: 10.68 vs. 74.89, SD: 10.33. In the group of patients with PCOS, infertility, acne, insulin resistance, and hyperandrogenism were significantly more common compared to the control group (Table 2).

The values of the tested hormone concentrations are presented in Table 3.

T and A concentrations were significantly higher (*p* < 0.000) in patients with PCOS. The mean values of these hormones were, respectively, 0.49 ng/mL, SD: 0.21 vs. 0.31 ng/mL, SD: 0.15 and 3.57 ng/mL, SD: 1.2 vs. control group, where the mean value was 2.15 ng/mL, SD: 0.84. The FAI value was also significantly higher in the study group than in the control: mean 1.09, SD: 0.59 vs. mean value: 0.89, SD: 0.73 (*p* < 0.029). The average value of total cholesterol in the study groups was 162.36 mg/dl, SD: 38.49 while that in the control group was 183.25 mg/dl, SD: 36.62. These differences were significant. The other lipid values between the groups did not differ significantly. 

We evaluated the OS parameters in the study and control group. The values of the OS parameters are presented in Table 4.

The values of the CAT activity in the PCOS group were considerably higher (370.76, SD: 70.31 vs. 220.19, SD 82.45). The MDA values between the control group and PCOS were similar, but these differences were significant. Due to the small sample size, this result may be accidental (0.10, SD: 0.02 vs. 0.09, SD: 0.03). The values of the other parameters of OS did not differ significantly between the study and the control group (Table 4). Correlations between the OS parameters and TChol, HDL, LDL, and TG in the PCOS and control group were examined (Table 5).

In the group of patients with PCOS, there were significant correlations between TChol and GPx activity (R = 0.59; *p* = 0.001), LDL-c and GPx activity (R = 0.43; *p* = 0.021), and TG and GPx activity (R = 0.5; *p* = 0.006) (Table 5). In the control group, there were significant reverse correlations between TChol and MDA concentrations (R = −0.4; *p* = 0.038) and LDL-c and MDA concentrations (R = −0.5; *p* = 0.008) (Table 5).

The values of the AIP, Castelli index, HOMA, and atherosclerosis risk index in both groups were also examined (Table 6). The average rates of the AIP and Castelli index did not differ significantly. The average HOMA values were higher in the PCOS group, with a trend close to significance. Patients with PCOS were divided into groups depending on the concomitant insulin resistance, hyperandrogenemia, and overweight and obesity status.

Among patients with PCOS, we examined whether IR, HA, or obesity and overweight affected the values of anthropometric parameters or the values of the AIP and Castelli index.

The values of parameters such as weight, HIP, waist circumference, BMI, and WHR were significantly higher in the group of patients with PCOS and IR (Appendix A).

The values of the studied parameters in the group of PCOS patients with and without hyperandrogenism did not differ significantly (Appendix A).

The values of the parameters studied, except for age and height, in the group of patients with PCOS and coexisting overweight and obesity differed significantly compared to the patients with PCOS and normal body weights (Appendix A).

The AIP and Castelli index values in the PCOS group with IR, hyperandrogenism, overweight, and obesity were compared to those in the PCOS group without IR and hyperandrogenism and with normal weight (Table 7). The values of the AIP and Castelli index in the PCOS group were significantly higher in the case of coexisting IR. The AIP and Castelli index values did not differ significantly in the group of patients with PCOS and HA compared to those in the group of patients without concomitant HA. For obese PCOS patients, the differences were not significant compared to normal-weight PCOS patients, but a trend was found for the Castelli index.

Correlations between the OS parameters and AIP and Castelli index in both groups of patients were also examined (Table 8). A significant correlation was found between the AIP and Castelli index and GPx activity in the PCOS group (Table 8).

A significant correlation was found between AIP and WHR, TChol, all cholesterol fractions, and systolic pressure in the PCOS group. A significant correlation was also present between the Castelli index and BMI, HDL-c, TG, SHBG, and systolic pressure in the PCOS group (Table 9).

In the control group, a significant correlation was found between the AIP and Castelli index and BMI, TChol, and all cholesterol fractions. Additionally, correlations were found between the Castelli index and WHR (Table 9).

## 4. Discussion

Patients with PCOS belong to an at-risk group for cardiovascular diseases. It is believed that the cause of this risk is overweight and obesity and the associated metabolic syndrome, hyperinsulinemia, IR, and hyperandrogenism. In this group of women, elevated values of oxidative stress parameters were also found [23,40]. PCOS women are also more likely to be diagnosed with cardiometabolic risk factors such as hypertension (HT) and type 2 diabetes (T2D). They also have more adverse lipid profiles compared to women without PCOS. Dyslipidemia is one of the risk factors for atherosclerosis. In our study, we found no significant differences in the lipid concentrations between the two groups (Table 3). Significantly higher TChol concentrations were found only in the control group. This result contrasts with the studies of other authors, who found significantly higher TChol and LDL values in patients with PCOS. Similar to other authors, we found no significant differences in the TG values in PCOS patients compared to a healthy control [41], perhaps because lipid disorders are often associated with overweight and obesity. In our study group, such patients accounted for less than 50% of the total while in the control group, they comprised less than 40%. The similar lipid profiles between PCOS and control patients may be due to the lack of significant differences between the groups in parameters such as body weight, BMI, and WHR.

Insulin resistance plays a major role in the development of metabolic syndrome in women with PCOS [14,42] and has also been implicated in the occurrence of hypertension in PCOS women through increased sodium reabsorption, and a reduction in the production of nitric oxide [14,43]. In our study, more IR patients were found in the PCOS group with values close to statistical significance (trend) (Table 6). However, due to the small sample size, this result may be accidental and could pose problems for the interpretation. In our study, in a group of patients with PCOS and concomitant IR, BMI, and WHR, the waist and hip circumference were significantly higher compared to PCOS patients without IR (Appendix A). Similar results were obtained in the PCOS group with concomitant overweight and obesity (Appendix A). In addition, in this group of patients, we observed significantly higher values of systolic and diastolic blood pressure.

Hyperandrogenemia is another risk factor for vascular disorders [44]. In our study, we found significantly higher androgen and FAI values in the PCOS group compared to the control group (Table 3). Due to the small sample size, however, this result should be treated with caution. HA, obesity (particularly abdominal obesity), and IR are characteristics of classic PCOS (phenotypes A and B). Patients with classic PCOS have more severe risk factors for T2DM and CVD [14]. Due to the high risk of cardiovascular complications in this group of women, early implementation of diagnostics and prevention is important. In the PCOS group with hyperandrogenism, we did not observe significant differences in the anthropometric parameters compared to patients with PCOS without coexisting hyperandrogenism.

Oxygen free radicals and OS are factors contributing to the development of atherosclerosis [45,46]. In our analyses, the MDA values and CAT activity were significantly higher in PCOS patients compared to the values in the control group (Table 4). MDA is a good biomarker of OS [24]. Some authors observed that the MDA values were significantly higher in PCOS patients with obesity [47] while others suggested that elevated values of MDA occur independently of obesity but correlate with IR [48,49]. In our study, the MDA values between the control group and PCOS were similar, but these differences were significant. Due to the small sample size, this result may be accidental and should be treated with caution. We examined the dependence of OS parameters on lipid values in both groups (Table 5). In the control group, we observed that MDA significantly correlated with TChol and LDL values. In the study group, we observed significant correlations of another OS marker, GPx, with the values of TChol, LDL, and TG. We did not observe any similar dependencies in the control group. CAT is an enzyme whose activity increases when disorders are preceded by inflammation. In our study, there was an increase observed in the CAT activity in the group with PCOS (Table 4). However, no significant correlations with the AIP and Castelli index were observed (Table 8). GPx is a macromolecular antioxidant (enzyme) [50]. This enzyme protects the organism by reducing lipid hydroperoxides to their corresponding alcohols and reducing H_2_0_2_ to water. In our work, GPx activity was not found to be significantly different between the study and control groups (Table 4). Some authors reported similar results and demonstrated that GPx did not differ between the PCOS group and a healthy control [51]. On the contrary, other authors observed lower GPx concentrations in PCOS patients [44]. Such divergent results between different studies may confirm that the group of patients with PCOS is heterogeneous. The obtained results may also be influenced by the different phenotypes of patients included in the study, the type of concomitant disorders, and other parameters such as age and BMI. On the other hand, differences between the group of patients with PCOS and the control may indicate slightly different mechanisms causing metabolic disorders and OS. Both of our groups showed no significant differences in age, weight, BMI, or WHR (Table 1). However, significant differences were observed in the hyperandrogenemia and IR values (Table 2), as both of these parameters were significantly higher in the PCOS group. This result may confirm that IR and androgen are factors that cause the different pathomechanisms of metabolic disorders and OS in patients with PCOS. Some authors investigated the OS parameters in PCOS patients with and without accompanying metabolic syndrome. Metabolic syndrome due to dyslipidemia reduces antioxidant activity [52].

We observed a significant relationship between the GPx, AIP, and Castelli index values but only in the PCOS group (Table 8). GPx may be active at different stages of development of a pathological condition in PCOS. It was suggested that in the case of PCOS, there are additional factors affecting the development of OS and a higher risk of cardiovascular complications. GPx, which has an affinity for hydrogen peroxide and organic hydroperoxides, is dependent on selenium, which in the Polish diet is usually deficient; this deficiency may intensify the adverse changes caused by the activities of free radicals [44]. Karolkiewicz et al. showed significant correlations between BMI and the atherosclerosis risk index in overweight patients [53]. The authors suggested that obesity reduces cytoprotective enzyme activity and causes oxidative stress [47,54]. Obesity is also a cause of chronic systemic inflammation. This activity manifests through increased serum levels of inflammatory cytokines and altered functions of peripheral blood lymphocytes. These changes might be responsible for the comorbidities often related to obesity, including diabetes, atherosclerosis, and steatohepatitis. The relationship between inflammation and the metabolic system is, moreover, found at the cellular level because macrophages and adipocytes are closely related. Inflammatory processes together with insulin resistance and OS are important risk factors for the development of CVD [55]. Obesity, especially abdominal obesity whose exponent is WHR, is considered one of the risk factors for cardiovascular diseases. In our analyses of patients with PCOS, a significant correlation was observed between the Castelli index and BMI and between both atherogenic factors and WHR (Table 9). Significant correlations were also found in the control group between atherogenicity indicators and BMI. Similar relationships were not found for WHR.

AIP is a novel index composed of HDL-c and TG and is used as an indicator of dyslipidemia and associated diseases (e.g., CVD). Zhu et al. found a strong association of AIP values with obesity [12]. In our study, the AIP values were significantly higher in the PCOS group with IR than in the control group (Table 7). The coexistence of overweight and obesity or hyperandrogenism in patients with PCOS did not increase the AIP value, and the value of the Castelli index in PCOS IR was insignificantly higher than that in the control. The increased value indicators of atherogenicity indicate that IR plays a special, negative role in the development of atherosclerosis. In our study, we found significant correlations between AIP and lipid concentrations, WHR, and systolic pressure (Table 9). Similar results were obtained for the Castelli index, with an additional significant correlation observed with BMI. In the control group, we observed similarly significant correlations between AIP and systolic pressure, HDL, and TG, and significant correlations with SHBG and BMI. In the case of the Castelli index, significant correlations were found for BMI, lipids, and androstendione. In our study, similar to Zhu et al., we showed a strong association of AIP with BMI in both groups and a correlation with WHR (an indicator of abdominal obesity) only in the PCOS group. It seems that the use of atherogenic indicators (AIP and Castelli index) may contribute to better and earlier identification of patients at high risk of CVD, especially among those with PCOS and IR. Notably, patients with PCOS most often report to gynecologists due to menstrual disorders, symptoms of hyperandrogenism, and infertility. Therefore, gynecologists should also pay attention to the risk factors for CVD in this group of women.

Our study confirms that PCOS with oxidative stress markers is a predisposing factor for the occurrence of cardiovascular complications. Strategies for the therapeutic management of patients with PCOS should be based not only on the treatment of infertility and menstrual disorders but also on the treatment of metabolic disorders, including hyperinsulinemia and hyperandrogenism, and the reduction in oxidative stress by improving antioxidative defense. These goals can be partly achieved through dietary therapy, exercise, and pharmacological antioxidants [23].

## 5. Strengths and Limitations

We found individual works evaluating atherogenicity indicators and measured a wide panel of parameters. The results of our study could have clinical applications and be useful in the treatment of patients with PCOS. However, the small size of the study sample may be a research limitation. Due to the small number of participants in the study and control groups, conclusions must be drawn with caution.

## 6. Conclusions

The severity of oxidative stress in women with PCOS correlates with exposure to cardiovascular diseases. Assessment of additional CVD parameters was found to be useful in identifying the risk groups for cardiometabolic disease among PCOS patients. In the case of additional risk factors such as hyperandrogenism and IR, for patients with PCOS, it is reasonable to include preventive examinations early. It is also important to evaluate lipidograms, which will make it possible to determine the indicators of atherogenicity. Patients with PCOS and IR are at particular risk of developing cardiovascular complications. PCOS should be considered an important risk factor for CVD, which is independent of the occurrence of obesity. This risk factor is related to the important role of insulin resistance, which also occurs independently of obesity.

Atherogenic factors (AIP and Castelli index) are useful additional parameters to assess the risk of cardiometabolic disease in PCOS patients, especially among those with insulin resistance. Early detection of risk factors should be an integral part of the care of PCOS patients. In laboratory studies on women with PCOS, TG, TChol, HDL-c and LDL-c levels, and GPx activity most clearly correlated with exposure to cardiovascular disease.

## Figures and Tables

**Table 1 biomedicines-10-01700-t001:** The results of anthropometric studies in the groups with PCOS and the control.

	Study Group *n* = 29		Control Group *n* = 27		*p*-Value
	Mean	SD	Mean	SD	
**Height (m)**	1.67	0.06	1.69	0.05	0.494
**Body weight (kg)**	74.24	19.22	69.56	13.77	0.475
**WC (cm)**	86.00	18.37	83.04	13.50	0.889
**HC (cm)**	104.83	11.33	100.52	11.38	0.084
**WHR**	0.82	0.14	0.83	0.13	0.210
**BMI (kg/m^2^)**	26.63	6.90	24.49	4.94	0.237

PCOS—polycystic ovary syndrome; WC—waist circumference; HC—hip circumference; BMI—body mass index; WHR waist–hip ratio; SD—standard deviation.

**Table 2 biomedicines-10-01700-t002:** Characteristics of the studied groups in terms of the occurrence of acne, seborrhea, infertility, insulin resistance, hyperandrogenism, overweight, and obesity in the PCOS group and control group.

	PCOS Group (*n* = 29)		Control Group (*n* = 27)		*p*-Value
	n	%	n	%	
**Acne**	16	55.1	1	3.7	0.001
**Seborrhea**	4	13.7	1	3.7	0.185
**Infertility**	7	24.1	1	3.7	0.029
**IR**	8	27.6	2	7.4	0.048
**Hyperandrogenism**	14	48.3	2	7.4	0.001
**Overweight and obesity**	14	48.3	10	37.1	0.396

IR—insulin resistance; PCOS—polycystic ovary syndrome.

**Table 3 biomedicines-10-01700-t003:** Values of glucose, hormone concentrations, systolic and diastolic pressure, TChol, HDL-c, LDL-c, and TGs of the blood in the group with PCOS and the control group.

	Study Group *n* = 29		Control Group *n* = 27		*p*-Value
	Mean	SD	Mean	SD	
**Glucose (G_0_) [mg/dL]**	87.68	9.77	85.49	8.84	0.471
**Insulin (I_0_) [mU/L]**	20.74	29.02	9.26	4.80	0.099
**T [ng/mL]**	0.49	0.21	0.31	0.15	2 × 10^−6^
**SHBG [nmol/L]**	52.44	24.53	50.41	24.77	0.623
**A [ng/mL]**	3.57	1.20	2.15	0.84	1 × 10^−5^
**FAI**	1.09	0.59	0.89	0.73	0.029
**Systolic pressure [mmHg]**	118.79	14.46	110.56	14.16	0.494
**Diastolic pressure [mmHg]**	78.17	10.68	74.89	10.33	0.475
**TChol [mg/dL]**	162.36	38.49	183.25	36.62	0.013
**HDL-c [mg/dL]**	60.29	15.57	66.90	14.78	0.090
**LDL-c [mg/dL]**	102.74	38.52	107.51	37.52	0.337
**TG [mg/dL]**	96.42	84.28	96.38	74.84	0.617

PCOS—polycystic ovary syndrome; SD—standard deviation, T—testosterone, SHBG—sex hormone-binding globulin, A—androstendione, FAI—free androgen index; TChol—total cholesterol; HDL-c high-density lipoprotein cholesterol; LDL-c low-density lipoprotein cholesterol; TGs—triglycerides.

**Table 4 biomedicines-10-01700-t004:** Parameters of oxidative stress in the group with PCOS and the control group.

	Study Group *n* = 29		Control Group *n* = 27		*p*
	Mean	SD	Mean	SD	
**MDA [** **µM]**	0.10	0.02	0.09	0.03	0.010
**CAT [k/gHb]**	370.76	70.31	220.19	82.45	2 × 10^−6^
**SOD [A/gHb]**	1683.6	268.8	1657.7	261.6	0.577
**GPx [U/gHb]**	4.95	0.68	4.91	0.74	0.712

MDA—malondialdehyde; CAT—catalase; SOD—superoxide dismutase; GPx—gluthation peroxidase.

**Table 5 biomedicines-10-01700-t005:** Spearman R correlation coefficients between OS and TChol, HDL-C, LDL-C, and TG in the PCOS group (*n* = 29) and control group (*n* = 27).

PCOS Group (*n* = 29)								
	MDA		CAT		SOD		GPx	
	R	*p*	R	*p*	R	*p*	R	*p*
**TChol [mg/dL]**	−0.10	0.600	−0.10	0.600	0.06	0.749	0.59	0.001
**HDL-c [mg/dL]**	−0.12	0.537	−0.10	0.608	0.12	0.530	−0.26	0.165
**LDL-c [mg/dL]**	−0.01	0.977	0.10	0.609	−0.16	0.412	0.43	0.021
**TG [mg/dL]**	0.19	0.314	0.25	0.194	−0.34	0.070	0.50	0.006
**Control Group (*n* = 27)**								
	**R**	** *p* **	**R**	** *p* **	**R**	** *p* **	**R**	** *p* **
**TChol [mg/dL]**	−0.40	0.038	−0.25	0.202	−0.24	0.224	0.29	0.137
**HDL-c [mg/dL]**	0.09	0.665	−0.16	0.429	−0.21	0.297	−0.01	0.964
**LDL-c [mg/dL]**	−0.50	0.008	−0.08	0.685	−0.13	0.520	0.32	0.108
**TG [mg/dL]**	−0.11	0.592	−0.01	0.959	0.21	0.304	0.17	0.402

TChol—total cholesterol; HDL-c high-density lipoprotein cholesterol; LDL-c low-density lipoprotein cholesterol; TGs—triglycerides; MDA—malondialdehyde; CAT—catalase; SOD—superoxide dismutase; GPx—gluthation peroxidase.

**Table 6 biomedicines-10-01700-t006:** Characteristics of the values of the indicators, AIP, Castelli index, and HOMA in the PCOS group (*n* =29) and control group (*n* = 27).

	Study Group (*n* = 29)		Control Group (*n* = 27)		*p*-Value
	Mean	SD	Mean	SD	
**AIP**	0.13	0.34	0.10	0.28	0.818
**Castelli index**	3.01	1.61	2.91	1.06	0.461
**HOMA**	4.51	6.25	1.99	1.13	0.094

AIP—atherogenic index of plasma; HOMA—homeostatic model assessment.

**Table 7 biomedicines-10-01700-t007:** Comparison of the Castelli index and AIP between patients in the PCOS group.

A.	PCOS and IR (*n* = 8)		PCOS without IR (*n* = 21)		*p*-Value
	Mean	SD	Mean	SD	
**AIP**	0.40	0.42	0.03	0.24	0.009
**Castelli index**	4.08	2.15	2.60	1.17	0.026
**B.**	**PCOS and HA (*n* = 14)**		**PCOS without HA (*n* = 15)**		
	**Mean**	**SD**	**Mean**	**Mean**	***p*-Value**
**AIP**	0.13	0.23	AIP	0.13	0.616
**Castelli index**	2.94	1.43	Castelli index	2.94	0.810
**C.**	**overweight and obesity PCOS group (*n* = 14)**		**Control group (*n* = 15)**		***p*-Value**
	**Mean**	**SD**	**Mean**	**SD**	
**AIP**	0.20	0.44	0.07	0.20	0.913
**Castelli index**	3.70	2.05	2.36	0.59	0.085

AIP—atherogenic index of plasma; PCOS—polycystic ovary syndrome.

**Table 8 biomedicines-10-01700-t008:** Spearman R correlation coefficients between OS parameters and the AIP and Castelli index in the PCOS group (*n* = 29) and control group (*n = 27*).

PCOS Group	MDA		CAT		SOD		GPX	
	R	*p*	R	*p*	R	*p*	R	*p*
**AIP**	0.19	0.326	0.22	0.250	−0.31	0.099	0.50	0.005
**Castelli**	0.13	0.505	0.10	0.592	−0.12	0.547	0.44	0.017
**Control Group**								
**AIP**	−0.17	0.384	0.12	0.566	0.33	0.098	0.13	0.506
**Castelli index**	−0.30	0.133	−0.03	0.877	−0.01	0.975	0.25	0.213

PCOS—polycystic ovary syndrome; MDA—malondialdehyde; CAT—catalase; SOD—superoxide dismutase; GPx—gluthation peroxidase; AIP—atherogenic index of plasma.

**Table 9 biomedicines-10-01700-t009:** Spearman R correlation coefficients between the AIP, Castelli index, BMI and WHR, cholesterol, blood pressure, and hormones in the PCOS group and control group.

	PCOS Group (*n* = 29)				Control Group (*n* = 27)			
PCOS Group	AIP		Castelli Index		AIP		Castelli Index	
	R	*p*	R	*p*	R	*p*	R	*p*
**BMI**	0.10	0.590	0.39	0.046	0.56	0.002	0.38	0.044
**WHR**	0.38	0.040	0.23	0.241	0.29	0.135	0.46	0.012
**TChol [mg/dL]**	0.42	0.022	0.06	0.758	0.63	0.000	0.65	0.000
**HDL-c [mg/dL]**	−0.77	0.000	−0.66	0.000	−0.66	0.000	−0.87	0.000
**LDL-c [mg/dL]**	0.63	0.000	0.18	0.356	0.74	0.000	0.80	0.000
**TG [mg/dL]**	0.94	0.000	0.86	0.000	0.42	0.030	0.72	0.000
**T [ng/mL]**	0.11	0.564	−0.12	0.556	−0.30	0.124	0.17	0.389
**SHBG [nmol/L]**	−0.20	0.297	−0.55	0.003	−0.34	0.085	−0.13	0.487
**A [ng/mL]**	0.10	0.600	−0.22	0.271	−0.55	0.003	0.07	0.706
**Systolic pressure (mmHg)**	0.37	0.047	0.48	0.011	0.27	0.167	0.28	0.137
**Diastolic pressure (mmHg)**	0.29	0.130	0.25	0.213	0.17	0.403	0.16	0.414

PCOS—polycystic ovary syndrome; BMI—body mass index; WHR—waist–hip ratio; Tchol—total cholesterol; HDL-c—high-density lipoprotein cholesterol; LDL-c—low-density lipoprotein cholesterol; TGs—triglycerides; A—androstendione; T—testosterone; SHBG—sex hormone-binding globulin; AIP—atherogenic index of plasma.

## Data Availability

The data will be made available on request.

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
