# Peer review of "The Role of Oxidative Stress in the Risk of Cardiovascular Disease and Identification of Risk Factors Using AIP and Castelli Atherogenicity Indicators in Patients with PCOS"

_biomedicines, 2022, doi:10.3390/biomedicines10071700_

Round 1

Reviewer 1 Report

This manuscript examined a group of women with Polycystic Ovarian Syndrome (PCOS) for markers of oxidative stress, insulin resistance, BMI, and hyperandrogenemia and assessed how these markers correlated with risk factors for developing atherosclerosis as measured with the Atherogenic Index of Plasma (AIP) and Castelli’s Index. The authors report correlations with a number of these factors with possible cardiovascular disease as indicated by AIP and/or Castelli’s index.

 Critique:

1.       The major critique of this manuscript is highlighted by the authors in the Strengths and limitations section where they note the small sample size. This problem is exacerbated when the PCOS group is broken into subgroups. These small n’s make interpretation difficult.

2.       The use of acronyms and initialisms is difficult to follow. Many are not clearly defined until later in the manuscript, if at all. These should be reviewed and clearly defined, perhaps in a table.

3.       Some differences are difficult to reconcile. For example, in table 4 the authors state the MDA in the PCOS group is considerably higher than controls, but the values are 0.10 vs 0.09. It is difficult to imagine how this difference is biologically significant. Total cholesterol in lower in the PCOS group, but there is little discussion of this. However, there is discussion of a statistically non-significant trend in HOMA values in table 6 that fits with the author’s hypothesis. These anomalies may be related to the low number of subjects.

4.       The authors should provide additional discussion of the relationships between adipose mass, oxidative stress and cardiovascular disease as these seem to have stronger correlations than PCOS.

Minor:

1.       Line 51: condition should be conditions

2.       Line 55: reduces should be reduce

3.       Line 61: occurrence diseases should be occurrence of diseases

4.       Line 62: there should be a comma after CVD

5.       Line 165: should read – equal to or greater than…

6.       Line 234: these hormone should be these hormones

7.       Lines 246-247: the sentence starting Values of MDA concentration… should be rewritten

8.       Table 5 is not properly aligned making it difficult to read.

9.       Line 317: remove he at the end of the line

1.   Line 358: protect should be protects

1.   Line 374: reduce should be reduces

1.   Line 376: should start—We observed a significant…

1.   Line 387: sentence should start—In our studies in patients with PCOS a significant…

1.   Line 394: Coexisted should be Coexisting

1.   Line 416: hyperinsulinemia is misspelled

1.   Line 418: there should be an and before pharmacological

1.   Line 441: clear should be clearly

1.   In the supplemental tables hight should be height

The problem being addressed in this manuscript is interesting and important. However, the small sample size makes interpretation difficult. The numbers should be increased. Also, the possible differences in adipose mass that could contribute to cardiovascular disease should receive additional dicussion.

Author Response

 Dear Reviewer,

thank you for your valuable comments.

Below I am attaching corrections and replies to your suggestions.

All suggested information has been incorporated into the text (in green colour).

 Critique:

1.       The major critique of this manuscript is highlighted by the authors in the Strengths and limitations section where they note the small sample size. This problem is exacerbated when the PCOS group is broken into subgroups. These small n’s make interpretation difficult.

We agree with the reviewer's opinion that the number of groups is small, which is emphasized in subgroups. Therefore, in the future, we plan to increase the sample size and continue research.

2.       The use of acronyms and initialisms is difficult to follow. Many are not clearly defined until later in the manuscript, if at all. These should be reviewed and clearly defined, perhaps in a table.

Suplement 4. Abbreviations

acronym/initialism

explanation

PCOS

polycystic ovary syndrome

IR

insulin resistance

CVD

cardiovascular disease

AIP

Atherogenic Index of Plasma

CAT

catalase

MDA

malondialdehyde

SOD

superoxide dismutase

GPx

Glutathione Peroxidase

NAFLD

non-alcoholic fatty liver disease

MetS

metabolic syndrome

HDL-c

high-density lipoprotein cholesterol fraction

LDL-c

low density lipoprotein cholesterol fraction

TChol

serum total cholesterol

TG

triglycerides

WHR

waist-hip ratio

BMI

body mass index

IHD

ischemic heart disease

CAD

coronary artery disease

LH

luteinizing hormone

FSH

follicle-stimulating hormone

SHBG

sex hormone binding globulin

OS

oxidative stress

DNA

deoxyribonucleic acid

AACE

American Association of Clinical Endocrinologists

ACE

American College of Endocrinology

T

testosterone

A

androstendione

I

insulin

EDTA

ethylendiaminetetraacetic acid

FAI

free androgen index

HOMA

Homeostatic Model Assessment

HPLC

high-performance liquid chromatography

WC

waist circumference

HC

hip circumference

HT

hipertension

T2DM

type 2 diabetes mellitus

3.       Some differences are difficult to reconcile. For example, in table 4 the authors state the MDA in the PCOS group is considerably higher than controls, but the values are 0.10 vs 0.09. It is difficult to imagine how this difference is biologically significant.

The MDA values between the control group and PCOS were similar, but these differences were significant. Due to the fact that the small sample size, this result may be accidental and should be treated with caution. Therefore, it will not be taken into account in our work.

This information has been incorporated into the text in Disscusion section.

Total cholesterol in lower in the PCOS group, but there is little discussion of this.

However, there is discussion of a statistically non-significant trend in HOMA values in table 6 that fits with the author’s hypothesis. These anomalies may be related to the low number of subjects.

Due to the small sample size, this result may be accidental and causes problems with interpretation. This information has been incorporated into the text in Disscusion section. The sentence has been reworded.

4.       The authors should provide additional discussion of the relationships between adipose mass, oxidative stress and cardiovascular disease as these seem to have stronger correlations than PCOS.

 Obesity is a cause of chronic systemic inflammation. This is manifested by increase serum levels of inflammatory cytokines and altered functions of peripheral blood lymphocytes. These changes might be responsible for comorbidities often related to obesity: diabetes and atherosclerosis. The relationship between inflammation and the metabolic system is found at the cellular level because macrophages and adipocytes are closely related. Inflammatory processes together with insulin resistance and OS are important risk factor for development of CVD [55].- it has been incorporated into the tekst.

Minor:

1.       Line 51: condition should be conditions – has been corrected

2.       Line 55: reduces should be reduce-has been corrected

3.       Line 61: occurrence diseases should be occurrence of diseases- has been corrected

4.       Line 62: there should be a comma after CVD- has been added

5.       Line 165: should read – equal to or greater than- has been corrected

6.       Line 234: these hormone should be these hormones- has been corrected

7.       Lines 246-247: the sentence starting Values of MDA concentration… should be rewritten: The MDA values between the control group and PCOS were similar, but these differences were significant. Due to the fact that the small sample size, this result may be accidental.- it is rewritten and incorporated into the text

8.       Table 5 is not properly aligned making it difficult to read- has been corrected.

9.       Line 317: remove he at the end of the line -has been removed

1.   Line 358: protect should be protects- has been corrected

1.   Line 374: reduce should be reduces- has been corrected

1.   Line 376: should start—We observed a significant…- has been corrected

1.   Line 387: sentence should start—In our studies in patients with PCOS a significant…- has been corrected

1.   Line 394: Coexisted should be Coexisting- has been corrected

1.   Line 416: hyperinsulinemia is misspelled- has been corrected

1.   Line 418: there should be an and before pharmacological- has been corrected

1.   Line 441: clear should be clearly- has been corrected

1.   In the supplemental tables hight should be height- has been corrected

The problem being addressed in this manuscript is interesting and important. However, the small sample size makes interpretation difficult. The numbers should be increased.

We agree with the reviewer's opinion that the number of groups is small, which is emphasized in subgroups. Therefore, in the future, we plan to increase the sample size and continue research.

Also, the possible differences in adipose mass that could contribute to cardiovascular disease should receive additional dicussion. In the disscusion section added the role of adipose tissue as a risk factor of CVD.

Reviewer 2 Report

Authors have determined the role of oxidative stress in the risk of cardiovascular disease and identified the risk factors using AIP and Castelli atherogenecity indicators in patients with PCOS. My comments are as under:

·         Although study design and outcomes are of interest but authors could have explained and prepare manuscript in much better way. There are several mistakes in the writing part and it was hard to understand the explanation in several sections of the manuscript.

·         In the ‘Discussions’ section of the manuscript the author must include the table no. for each parameter they are discussing about for the ease of comparing.

·         Line 246: How MDA is considerably higher in study groups as compared to control group (0.10 Vs 0.09); needs clarification.

·         Line 338: FAI values in study group did not differ much from that of control, hence can’t agree with the statement.

Spelling mistakes and typos:

·         Line 68: Dislipidemia to Dyslipidemia

·         Line 107 and throughout the manuscript:  Gluthation peroxidase (GPx) into Glutathione Peroxidase.

·         Line 108 and throughout the manuscript: malanodialdehyde to malondialdehyde.

·         Line 176: Full form of HA must be addressed.

·         Line 210: The sentence ‘> 8 points F-G’ is unclear.

·         Line 222: Correct Tab. 2 as Table 2 (also in line 292)

·         Line 233: Author has mentioned p< 0.000, but in table 3, it shows p-value is equal to 0.00, needs a clarification.

·         Line 262: API into AIP

·         Line 309: ‘a’ letter of the word additionally should be replaced with ‘A’.

·         Line 317:  the sentence should be corrected ‘Dyslipidemia is one of the he risk factor ’.

·         Line 28 : Full form for CV must be addressed.

·         Typos error :  7.4% (line 208) , 0.59 and 0.73 (p< 0.029)(Line 237), 36.62 (Line 239 ), AIP i castelli  index (Line 298), Karol… et al (Line 382), observed (Line 387), Zhu et al
(line 404). Referencing style should be uniform throughout the manuscript either et al., or just et al

·         Table 1 : Spelling of Height and its unit

·         Table 3: units for all the parameters need to be mentioned.

·         Table 4: Latin symbol for micromolar should be used (µ not u).

·         Table 5: Spelling error in HDL-c

Author Response

Dear Reviewer,

thank you for your valuable comments.

Below I am attaching corrections and replies to your suggestions.

All suggested information has been incorporated into the text (in red colour).

Comments and Suggestions for Authors

Authors have determined the role of oxidative stress in the risk of cardiovascular disease and identified the risk factors using AIP and Castelli atherogenecity indicators in patients with PCOS. My comments are as under:

·         Although study design and outcomes are of interest but authors could have explained and prepare manuscript in much better way. There are several mistakes in the writing part and it was hard to understand the explanation in several sections of the manuscript.

·         In the ‘Discussions’ section of the manuscript the author must include the table no. for each parameter they are discussing about for the ease of comparing. - The table no. have been included

·         Line 246: How MDA is considerably higher in study groups as compared to control group (0.10 Vs 0.09); needs clarification.-

The MDA values between the control group and PCOS were similar, but these differences were significant. Due to the fact that the small sample size, this result may be accidental and should be treated with caution. Therefore, it will not be taken into account in our work.

This information has been incorporated into the text in Disscusion section ( in green colour)

·         Line 338: FAI values in study group did not differ much from that of control, hence can’t agree with the statement.

In our study, FAI values in both groups were similar, but the differences were significant. However, due to the small size of the group, this result is difficult to interpret and should be treated with caution- this sentence has been incorporated into the text in Discussion section

Spelling mistakes and typos:

·         Line 68 (67): Dislipidemia to Dyslipidemia- has been corrected

·         Line 107 and throughout the manuscript:  Gluthation peroxidase (GPx) into Glutathione Peroxidase.- has been corrected

·         Line 108 and throughout the manuscript: malanodialdehyde to malondialdehyde- has been corrected.

·         Line 176: Full form of HA must be addressed- has been corrected.

·         Line 210: The sentence ‘> 8 points F-G’ is unclear- has been changed.

·         Line 222: Correct Tab. 2 as Table 2 (also in line 292)- has been corrected

·         Line 233: Author has mentioned p< 0.000, but in table 3, it shows p-value is equal to 0.00, needs a clarification. has been corrected

·         Line 262: API into AIP-has been corrected

·         Line 309: ‘a’ letter of the word additionally should be replaced with ‘A’.- has been corrected

·         Line 317:  the sentence should be corrected ‘Dyslipidemia is one of the he risk factor ’. has been corrected

·         Line 28: Full form for CV must be addressed- has been corrected.

·         Typos error :  7.4% (line 208) , 0.59 and 0.73 (p< 0.029)(Line 237- 243), 36.62 (Line 239 ), AIP i castelli  index (Line 298), Karol… et al (Line 382), observed (Line 387), Zhu et al- has been corrected
(line 404). Referencing style should be uniform throughout the manuscript either et al., or just et al-
has been corrected

·         Table 1 : Spelling of Height and its unit has been corrected

·         Table 3: units for all the parameters need to be mentioned. has been added

·         Table 4: Latin symbol for micromolar should be used (µ not u) has been corrected.

·         Table 5: Spelling error in HDL-c has been corrected

 The article has been  english edited by MDPI.

Round 2

Reviewer 1 Report

The manuscript is much improved. There is still and issue with the low numbers of participants, but the authors acknowledge this. The low numbers do contribute to the difficulties in interpreting results.

Reviewer 2 Report

No further comments